# Optimum Shifting to Stabilize Training and Improve Generalization of Deep Neural Networks

## Abstract

Recent studies have shown that the generalization of neural networks is correlated with the sharpness of the loss landscape and flat minima suggests a better generalization ability than sharp minima. In this paper, we introduce a method called optimum shifting (OS), which changes the parameters of a neural network from sharper minima to a flatter one while maintaining the same training loss. Our approach is based on the observation that when the input and output of a neural network are fixed, the matrix multiplications within the network can be treated as systems of under-determined linear equations, enabling adjustment of parameters in solution space. This can be accomplished by solving a constrained optimization problem, which is easy to implement. Furthermore, we introduce a practical stochastic optimum shifting (SOS) technique utilizing neural collapse theory to reduce computational costs and provide more degrees of freedom for optimum shifting. In our experiments, we present various DNNs (e.g., VGG, ResNet, DenseNet, and Vit) on the CIFAR 10/100 and Tiny-Imagenet datasets to validate the effectiveness of our method.

## 1 Introduction

Deep Neural Networks (DNNs) are powerful and have shown remarkable results in various fields, including computer vision (Goodfellow et al., 2020; Sohl-Dickstein et al., 2015; Kingma & Welling, 2013) and natural language processing (OpenAI, 2023; Vaswani et al., 2017). It formulates a learning problem as an optimization problem and utilizes stochastic gradient descent and its variants to minimize the loss function:

$$\min_{\Theta} \frac{1}{n} \sum_{i=1}^{n} L(f(\mathbf{x}_i, \Theta), \mathbf{y}_i). \tag{1}$$

Today, DNNs are overparameterized and capable of providing larger hypothesis space with normally better solutions having small training errors. However, this expansive hypothesis space is concurrently populated with different minima, each characterized by distinct generalization abilities. Recent studies have shown that generalization is correlated with the sharpness of the loss landscape and flat minima suggest a better generalization ability than sharp minima Keskar et al. (2016); Neyshabur (2017); Hochreiter & Schmidhuber (1994); Keskar et al. (2017); Chaudhari et al. (2019); Gatmiry et al. (2023). In this work, we aim at answering this question:

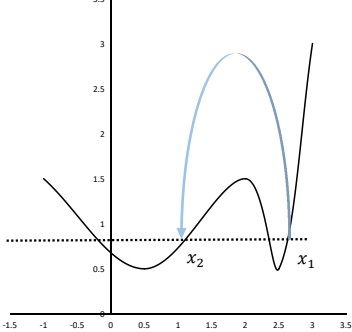

Figure 1: Schematic of Optimum Shifting

*Can we modify the parameters of a neural network from one point to a flatter one while maintaining the same training loss?*

In this paper, we propose a method called optimum shifting (OS) to attain this specific objective. It changes the parameters of a neural network from the current point to a flatter one while maintaining

the same training loss value. This approach is based on the matrix multiplication within the neural network:

$$\boldsymbol{A}\mathbf{v} = \boldsymbol{b}, \tag{2}$$

where $\boldsymbol{A} \in \mathbb{R}^{m \times n}$ represents the input matrix, and $\mathbf{v} \in \mathbb{R}^n$ denotes the parameters in the neural network's linear layer. Consider it to be a set of linear equations, and the parameter $\mathbf{v}$ can be modified in the solution space. Assume that the rows in $\boldsymbol{A}$ are independent without loss of generality, the equation $\boldsymbol{A}\mathbf{v} = \boldsymbol{b}$ is under-determined if $m < n$ and has infinite solutions for $\mathbf{v}$. This property allows us to move the neural network parameters from the current point $\mathbf{v}$ in the solution space to another point $\mathbf{v}^*$ by minimizing the sharpness (in particular, the trace of its Hessian (Gatmiry et al., 2023)), which can be calculated by solving a simple constraint optimization problem.

The main challenge for the method above is that it requires keeping training loss unchanged across all training samples. Using the entire training set for OS demands a significantly large memory capacity and huge computational costs. Moreover, OS in the whole dataset results in a less under-determined input matrix, which limits the degrees of freedom for optimum shifting. To overcome these challenges, we take inspiration from stochastic gradient descent, which uses a small batch to conduct gradient descent, thereby reducing computational costs. We propose stochastic optimum shifting, which performs OS on a small batch of data. With this approach, the generalization ability of the neural network is improved and computational costs are decreased. According to the theory of the Neural Collapse ($\mathcal{NC}$) (Zhu et al., 2021; Tirer & Bruna, 2022; Zhou et al., 2022), *if the loss is unchanged in a small batch of training data (typically, "$batch \geq n$" where n is the class number), it is expected to be unchanged across all training data.* Therefore, the empirical loss is expected to remain unchanged under stochastic optimum shifting algorithm.

To summarize, our contributions include:

- We propose optimum shifting (OS) and stochastic optimum shifting (SOS), which enable us to modify the parameters of neural networks while maintaining the same training loss value. We prove that the generalization ability has increased in the Hessian trace perspectives.

- We present experiments on two recognition tasks to verify the effectiveness of OS. We train VGG, ResNets, DenseNets, and Vit-B on CIFAR10/100 and Tiny-ImageNet datasets and Yolo detector on the PASCAL VOC dataset. Experiments show that by using OS, the training process can be stabilized and models can obtain better generalization.

- The proposed OS and SOS are compatible and can be easily integrated into traditional regularization techniques, such as weight decay and recently proposed approaches to find flatter minima, such as the sharpness-aware minimization (SAM) method (Foret et al., 2021).

## 2 RELATED WORK

**Flatness and Generalization** Research on the correlation between generalization and sharpness can be traced back to (Hochreiter & Schmidhuber, 1994). (Yao et al., 2020; Jiang et al., 2019) perform a large-scale empirical study on various notions of generalization measures and show that sharpness-based measures correlate with generalization best. (Keskar et al., 2017) observe that when increasing the batch size of SGD, the test error and sharpness of the trained model will both increase. (Gatmiry et al., 2023) show that with standard restricted isometry property on the measurement, minimizing the trace of Hessian can lead to better generalization. Although (Dinh et al., 2017) argue that for networks with scaling invariance, there always exist models with good generalization but with arbitrarily large sharpness. However, it does not contradict our main result here, which only asserts the solution with a minimal trace of Hessian generalizes well, but not vice versa. Therefore, recent studies have proposed several penalty-based sharpness regularization methods to improve the generalization. SAM (Foret et al., 2021) was proposed to penalize the sharpness of the landscape to improve the generalization. The full-batch SAM aims to minimize worst-direction sharpness (Hessian spectrum) and 1-SAM aims to minimize the average-direction sharpness (Hessian trace) (Wen et al., 2023). Furthermore, (Kwon et al., 2021) proposed adaptive SAM, where optimization could keep invariant to a specific weight-rescaling operation. In addition, (Zhao et al., 2022) proposes to improve the generalization by penalizing the gradient norm of loss function during optimization. It is worth noting that the methods above are all penalty-based methods, i.e. adding a penalty term, which represents the flatness, to the loss function. Compared with them, our method is a constraint and

objective separation method, which separates the flatness and loss value as two isolated objectives to optimize. As a result, our OS can be easily integrated into penalty-based methods (e.g. SAM) to achieve better performance.

**Neural Collapse.** Recent seminal works empirically demonstrated that last-layer features and classifiers of a trained DNN exhibit an intriguing Neural Collapse ($\mathcal{NC}$) phenomenon (Zhu et al., 2021; Tirer & Bruna, 2022; Zhou et al., 2022). Specifically, it has been shown that the learned features (the output of the penultimate layer) of within-class samples converge to their class means. Moreover, $\mathcal{NC}$ seems to take place regardless of the choice of loss functions. We utilize this phenomenon and propose stochastic optimum shifting, which can reduce the computational costs and provide more degrees of freedom for optimum shifting. For example, when we apply optimum shifting to ResNet/DenseNet for CIFAR 100 classification, by ensuring the loss remains unchanged across 100 samples, the loss remains probabilistic unchanged on the whole 50,000 training data points.

## 3 OPTIMUM SHIFTING

### 3.1 NOTATIONS

Throughout this paper, we denote $\mathcal{S} \triangleq \cup_{i=1}^n \{(\boldsymbol{x}_i, \boldsymbol{y}_i)\}$ as the training set containing $n$ training samples, $L$ as the loss function, and $f(\boldsymbol{x}_i)$ as the neural network approximation. A l-convolutional-layer neural network is expressed as:

$$f(\boldsymbol{x}) = \mathbf{v}^T \text{vec} \left\{ \sigma(\boldsymbol{F}_l * \sigma(\boldsymbol{F}_{l-1} * \cdots \sigma(\boldsymbol{F}_1 * \boldsymbol{x}) \cdots)) \right\}, \tag{3}$$

where $*$ denotes the convolution operator, $\sigma(x) = \max\{0, x\}$ is the entry-wise ReLU activation and $\boldsymbol{F}_l$ is the convolution in the $l$-th layer. The vectorized output of $i$-th layer is represented using $\text{vec}(\boldsymbol{x}_{l,i}) \in \mathbb{R}^{m_l}$. We vectorized parameters in each layer and stack it as $\mathbf{W} = [\text{vec}(\mathbf{v}), \text{vec}(\mathbf{F}_l), \cdots, \text{vec}(\mathbf{F}_1)]$

### 3.2 MAIN RESULTS

Before proceeding, we first define the flatness of neural networks. There are many measures to define the flatness. But currently, the trace of Hessian has been theoretically proved with the generalization bound (Gatmiry et al., 2023). In this paper, we define flatness by the Hessian trace.

**Definition 1** *The flatness $F(L)$ is defined as the trace of the Hessian matrix $\boldsymbol{H}_L$ of the loss function $L$ with respect to network parameters $\mathbf{W}$ :*

$$F(L) \triangleq \text{tr}(\boldsymbol{H}_L) = \text{tr}(\nabla_{\mathbf{W}}^2 L(\boldsymbol{f}(\boldsymbol{x}_i), \boldsymbol{y}_i)). \tag{4}$$

Next, we show the main theorem of our paper. It shows that for different neural networks such as CNN, ResNet, DenseNet and MLP, the lower bound and upper bound of the Hessian trace are linear with the Frobenius norm of the weight in the final linear layer. So if we minimize the Frobenius norm of the weight in the final linear layer, both the upper bound and lower bound of the Hessian trace will also be minimized, thus suggesting a better generalization ability.

**Theorem 1** *For a l-convolutional-layer neural network, given the loss function L, the trace of Hessian can be upper bounded by:*

$$C_0 + C_1 \|\mathbf{v}\|^2 \le tr(\boldsymbol{H}_L) \le C_0 + C_2 \|\mathbf{v}\|^2 \tag{5}$$

*where $C_0, C_1, C_2$ are constants and independent of the last hidden layer's weight $\mathbf{v}$. So if $\|\mathbf{v}\|^2$ is minimized, both the upper bound and lower bound of the Hessian trace will also be minimized.*

**Theorem 2** *For a l-convolutional-layer ResNet, given the loss function L, the trace of Hessian can be upper bounded by:*

$$C_0 + C_3 \|\mathbf{v}\|^2 \le tr(\boldsymbol{H}_L) \le C_0 + C_4 \|\mathbf{v}\|^2 \tag{6}$$

*where $C_0, C_3, C_4$ are constants and independent of the last hidden layer's weight $\mathbf{v}$. So if $\|\mathbf{v}\|^2$ is minimized, both the upper bound and lower bound of the Hessian trace will also be minimized.*

**Theorem 3** *For a l-convolutional-layer DenseNet, given the loss function L, the trace of Hessian can be upper bounded by:*

$$C_0 + C_5\|\mathbf{v}\|^2 \leq tr(\boldsymbol{H}_L) \leq C_0 + C_6\|\mathbf{v}\|^2 \tag{7}$$

*where $C_0, C_5, C_6$ are constants and independent of the last hidden layer's weight $\mathbf{v}$. So if $\|\mathbf{v}\|^2$ is minimized, both the upper bound and lower bound of the Hessian trace will also be minimized.*

**Theorem 4** *For a l-fully-connected neural network, given the loss function L, the trace of Hessian can be upper bounded by:*

$$C_0 + C_7\|\mathbf{v}\|^2 \leq tr(\boldsymbol{H}_L) \leq C_0 + C_8\|\mathbf{v}\|^2 \tag{8}$$

*where $C_0, C_7, C_8$ are constants and independent of the last hidden layer's weight $\mathbf{v}$. So if $\|\mathbf{v}\|^2$ is minimized, both the upper bound and lower bound of the Hessian trace will also be minimized.*

The proof of the four theorems is detailed in Appendices A.1 to A.4.

## 3.3 METHODOLOGY

The linear layer $\mathbb{R}^m \to \mathbb{R}^n$ with activation function $\sigma$ can be represented as follows:

$$\phi^{fc}(\mathbf{v}) = \sigma(\boldsymbol{A}\mathbf{v} + \boldsymbol{c}) \tag{9}$$

where $\boldsymbol{A} \in \mathbb{R}^{batch \times m}$, $\mathbf{v} \in \mathbb{R}^{m \times n}$, $\boldsymbol{c} \in \mathbb{R}^{batch \times n}$ and $batch$ is the input batch size. We denote the result of $\boldsymbol{A}\mathbf{v}$ as :

$$\boldsymbol{A}\mathbf{v} := \boldsymbol{b}. \tag{10}$$

The training loss value of a neural network will not change no matter how the parameter $\mathbf{v}$ is adjusted if the input matrix $\boldsymbol{A}$ and output matrix $\boldsymbol{b}$ are both fixed. Specifically, the equation $\boldsymbol{A}\mathbf{v} = \boldsymbol{b}$ defines a system of linear equations. If this system is under-determined, then $\mathbf{v}$ has an infinite number of solutions. Any option in the solution space is available as a replacement for the current $\mathbf{v}$. As stated in Section 3.2, both the upper bound and lower bound of the Hessian trace are linear with $\|\mathbf{v}\|^2$. If $\|\mathbf{v}\|^2$ is minimized, both the upper bound and lower bound of the Hessian trace will also be minimized. As a result, we prefer replacing the current point with the one that has the least Frobenius norm in the solution space. This can be obtained by solving a least-square problem as follows:

$$\text{minimize} \quad \|\mathbf{v}\|^2 \tag{11}$$
$$\text{subject to} \quad \boldsymbol{A}\mathbf{v} = \boldsymbol{b}. \tag{12}$$

Thus, we aim to find the point with the smallest Frobenius norm in the solution space to replace current $\mathbf{v}$. Because $\mathbf{v} \in \mathbb{R}^{m \times n}$ is a matrix, we need to decompose it into $n$ independent least-squares problems. The $i$-th column of $\mathbf{v}$ is denoted as $\mathbf{v}_i$, and the Lagrangian for the least-square problem is:

$$L_1(\mathbf{v}_1, \cdots, \mathbf{v}_n, \boldsymbol{\lambda}_1, \cdots, \boldsymbol{\lambda}_n) = \sum_{i=1}^{n}(\mathbf{v}_i^T\mathbf{v}_i + \boldsymbol{\lambda}_i^T(\boldsymbol{A}\mathbf{v}_i - \boldsymbol{b}_i)). \tag{13}$$

Since $L_1$ is a convex quadratic function of each $(\mathbf{v}_i, \boldsymbol{\lambda}_i)$, we can find the minimum $(\mathbf{v}_i^*, \boldsymbol{\lambda}_i^*)$ from the optimality condition:

$$\nabla_{\mathbf{v}_i}L_1 = 2\mathbf{v}_i + \boldsymbol{A}^T\boldsymbol{\lambda}_i = 0, \tag{14}$$
$$\nabla_{\boldsymbol{\lambda}_i}L_1 = \boldsymbol{A}\mathbf{v}_i - \boldsymbol{b}_i = 0, \tag{15}$$

which yeilds a closed form solution $\mathbf{v}_i^* = \boldsymbol{A}^T(\boldsymbol{A}\boldsymbol{A}^T)^{-1}\boldsymbol{b}_i$. By resolving $n$ independent least square problems, we can finally identify the point with the smallest Frobenius norm as $\mathbf{v}^* = [\mathbf{v}_1^*, \mathbf{v}_2^*, \cdots, \mathbf{v}_n^*] = \boldsymbol{A}^T(\boldsymbol{A}\boldsymbol{A}^T)^{-1}\boldsymbol{b}$.

## 4 STOCHASTIC OPTIMUM SHIFTING

As stated above, when the neural network's parameters is changed by OS, the training loss is expected to remain unchanged for all data samples. But for implementation, it is hampered by the three issues.

- **Computing large batch size:** The typical batch size for neural network training is much smaller than the dataset size. However, to perform OS, the neural network must be fed the entire dataset, which increases the computational costs and requires large memory size.

- **Performing Gaussian elimination:** The rows of the input matrix were presumed independent in the previous section. But this supposition isn't always true in actual circumstances. Gaussian elimination is utilized to reduce the linearly dependent row vectors. There is a sizable computational challenge involved in applying Gaussian elimination to a large matrix.

- **Degrees of freedom for optimum shifting:** When conducting optimum shifting, smaller rows and larger columns of the input matrix $A$ result in a larger solution space of the equation $Av = b$, increasing the degrees of freedom to find a flatter $v$. However, the rows of $A$ represent the batch size, which must be large in the first issue.

These problems lead to the question that: *How can we reduce the computational costs while providing additional degrees of freedom for optimum shifting?* For the first one, we take inspiration from stochastic gradient descent, which balances convergence efficiency and computational costs by performing gradient descent on a small batch. We propose a practical approach named stochastic optimum shifting (SOS) to reduce computational costs. Specifically, it uses a small batch to conduct OS after training the model for each $s$ iteration. The whole training process is as follows: First, to make the input matrix $A$ row independent, we perform Gaussian elimination to the linear system equations. Then we follow the method stated in Section. 3.3 to compute the new parameter. The algorithm details are outlined in Algorithm 1.

**Input:** Training set $\mathcal{S} \triangleq \cup_{i=1}^{n}\{(\boldsymbol{x}_i, \boldsymbol{y}_i)\}$, batch size $b_1, b_2$ for SGD and SOS, step size $\gamma > 0$.

**for** *number of training epochs* **do**
  Sample batch $\mathcal{B} = \{(\boldsymbol{x}_1, \boldsymbol{y}_1), ... (\boldsymbol{x}_{b_2}, \boldsymbol{y}_{b_2})\}$;
  Compute input and output matrix ;
  $A = \begin{bmatrix} \boldsymbol{x}_{L,1}, \boldsymbol{x}_{L,2}, \cdots, \boldsymbol{x}_{L,b_2} \end{bmatrix}$ $\boldsymbol{b} = \begin{bmatrix} \mathbf{v}^T \boldsymbol{x}_{L,1}, \mathbf{v}^T \boldsymbol{x}_{L,2}, \cdots, \mathbf{v}^T \boldsymbol{x}_{L,b_2} \end{bmatrix}$ ;
  **for** *each columns $\mathbf{v}_i$ in the final linear layer* **do**
    Gaussian elimination to make $A$ row independent:
    $[A_*, \boldsymbol{b}_{i*}] = Gaussian\ eliminate([A, \boldsymbol{b}_i])$ ;
    Update the parameters:
    $\mathbf{v}_i^* = A_*(A_*(A_*)^T)^{-1}\boldsymbol{b}_{i*}$;
  **end**
  **for** $t = 0, 1, \cdots, s$ **do**
    Update all parameters using SGD;
    $\mathbf{W}_t = \mathbf{W}_{t-1} - \gamma \frac{1}{b_1} \sum_{i=1}^{b_1} \nabla_{\mathbf{W}_{t-1}} L$;
  **end**
**end**

**Algorithm 1:** SOS algorithm during training

Figure 2: Schematic of the OS algorithm.

To achieve optimum shifting, the training loss is expected to remain unchanged across the entire dataset. However, for stochastic optimum shifting, we only maintain it unchanged in a small batch. As we will show in Section. 5.2.1, performing stochastic optimum shifting with a limited batch of 300 images scarcely increases the empirical training loss. Instead, it sometimes helps to stabilize the training process. A recent study in neural collapse (Zhu et al., 2021; Tirer & Bruna, 2022; Zhou et al., 2022) helps us to understand this phenomenon. It reveals that: as training progresses, the within-class variation of the activations becomes negligible as they collapse to their class means. For example, when training on CIFAR100 dataset, the images will converge to the 100 class means. Therefore, if the loss of 100 images remains unchanged after performing optimum shifting to the final fully connective layer, the loss of the entire dataset will also remain nearly unchanged.

Moreover, the small batch for optimum shifting also offers more degrees of freedom for optimum shifting. For instance, when the last layer maps a vector with 1024 dimensions to 100 dimensions. The weight matrix $\mathbf{v} \in \mathbb{R}^{1024 \times 100}$. When feeding the entire dataset to the neural network, the input matrix $A \in \mathbb{R}^{50000 \times 1024}$, which may not be under-determined. When using stochastic optimum

shifting, the input matrix $A \in \mathbb{R}^{300 \times 1024}$, which means that the systems of linear equations are under-determined. And it must have infinite solutions in the solution space.

## 5 EXPERIMENT VALIDATION

In order to assess SOS's efficacy, we investigate the performance of different DNNs (VGG, ResNet, DenseNet, ViT) on different computer vision tasks, including CIFAR10/100, Tiny-Imagenet classification and object detection. In the first subsection, SOS is applied to trained deep models, and in the second subsection, it is further applied repeatedly in the training process of deep models. When SOS is applied in the training process, we compare our method with two other training schemes: one is the standard SGD training scheme and the other is the SAM (Foret et al., 2021) training scheme. As we can see below, SOS improves the generalization ability of trained models, standard SGD scheme, and SAM training scheme.

Table 1: Test Accuracy on CIFAR Classification for trained models

| Dataset | CIFAR100 | | | | CIFAR10 | | | |
| Augmentation | Basic | | Mixup | | Basic | | Mixup | |
| SOS | ✓ | × | ✓ | × | ✓ | × | ✓ | × |
| VGG-16 | 70.23 | 70.18 | **73.86** | 72.74 | 91.81 | 91.76 | **93.43** | 93.12 |
| ResNet-18 | 77.68 | 77.53 | **80.99** | 79.11 | 95.12 | 94.96 | **95.93** | 95.66 |
| ResNet-50 | 78.12 | 77.85 | **81.54** | 81.33 | 95.11 | 95.08 | **96.41** | 96.03 |
| DenseNet-201 | 80.24 | 80.07 | **82.45** | 81.95 | 95.41 | 95.34 | **96.21** | 96.17 |

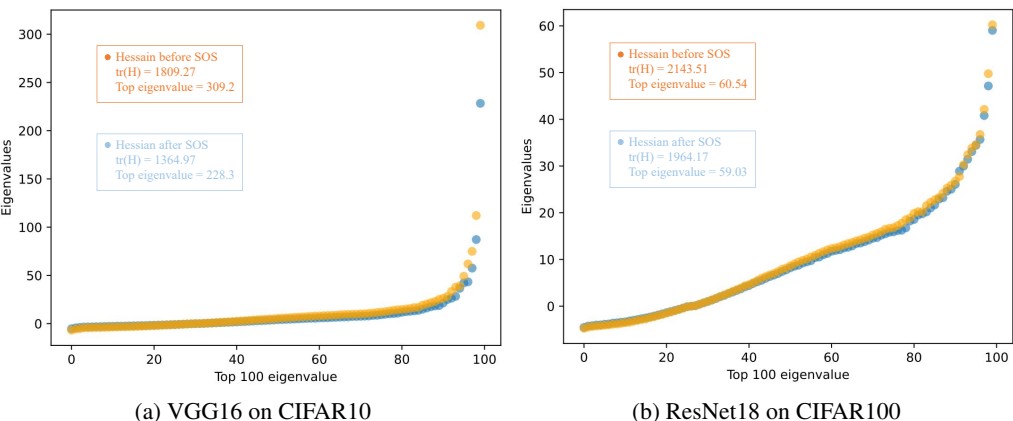

(a) VGG16 on CIFAR10       (b) ResNet18 on CIFAR100

Figure 3: Top 100 eigenvalues of the Hessian matrix before and after SOS.

### 5.1 APPLY SOS TO TRAINED MODELS

**Test Accuracy** We first evaluate SOS by applying it to trained deep models on CIFAR10 and CIFAR100 dataset (Krizhevsky et al., 2009), which consists of 50k training images and 10k testing images in 10 and 100 classes. Different convolutional neural network architectures are tested, including relatively simple architectures, such as VGG (Simonyan & Zisserman, 2014), and complex architectures, such as ResNet (He et al., 2016) and DenseNet (Huang et al., 2017). For the training datasets, we employ data augmentations. One is the basic augmentation (basic normalization and random horizontal flip) and the other is Mixup augmentation (Zhang et al., 2017). Table. 1 shows the result. We can see all the test accuracy has been improved slightly by SOS. For example, the test accuracy of VGG-16 on CIFAR100 has been improved from 72.74% to 73.86%.

**Hessian Analysis** We visualize the top 100 eigenvalues of the Hessian matrix before and after SOS in ascending order as shown in Figure 3. The orange and blue spots represent the eigenvalue before and after SOS. We also calculate the trace and top 1 eigenvalue for analysis. We can see that both the trace (average-direction sharpness) and the top 1 eigenvalue (worst-direction sharpness) have been minimized by SOS, which is consistent with our proposed Theorem 1.

## 5.2 APPLY SOS IN TRAINING PROCESS

The smoothness of loss landscape can benefit the optimization process of neural networks (Gouk et al., 2021; Santurkar et al., 2018). We argue that SOS can smooth the loss landscape, thus stabilizing the training process and improving the generalization ability of neural networks. In this section, we apply SOS during the training process and analysis the loss and test accuracy.

### 5.2.1 CIFAR10 AND CIFAR100 CLASSIFICATION

**Loss analysis.** We first analyze the training loss with and without SOS. We use SGD, Adam (Kingma & Ba, 2014), and Adagrad (Duchi et al., 2011) to train VGG16 on CIFAR10, ResNet18 (He et al., 2016) and DenseNet121 (Huang et al., 2017) on CIFAR100 (Krizhevsky et al., 2009) without any data augmentation. As shown in Fig. 4, the *x*-axis is the training epochs and the *y*-axis is the training loss value.

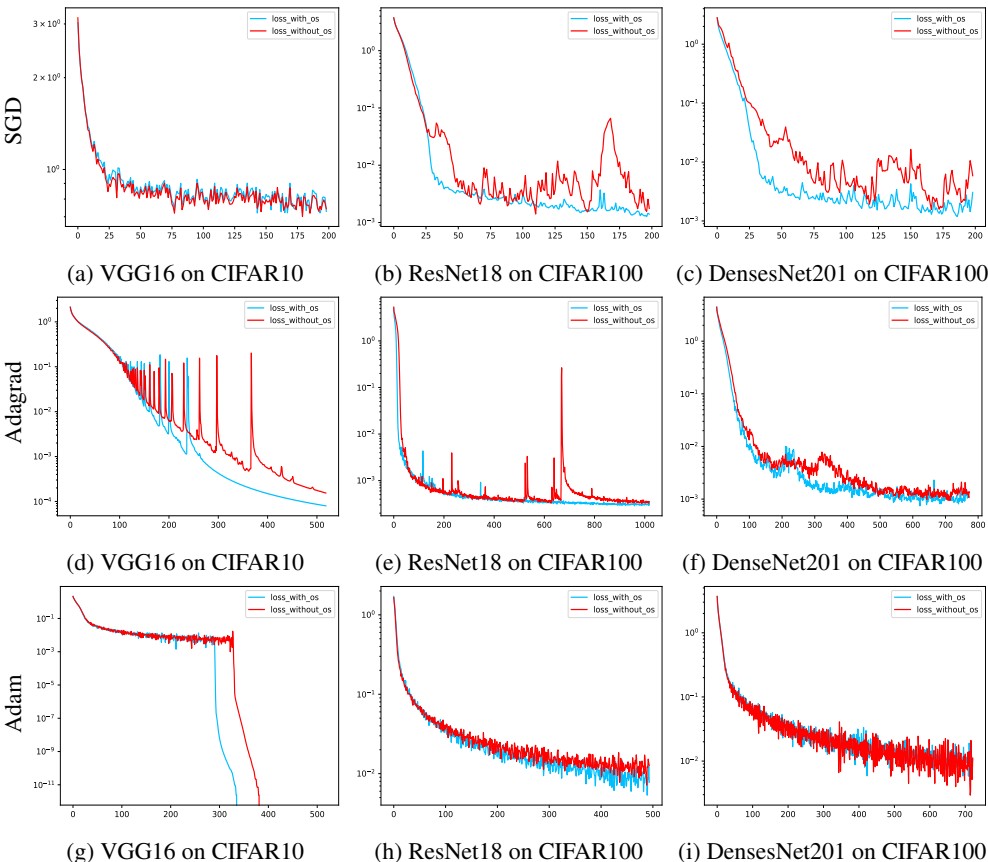

Figure 4: Visualization of the training loss for different models and optimizers with and without SOS.

Figure. 4 shows that the loss with SOS is almost the same or lower than the loss without SOS, which validates our statement in Section. 4 that SOS will not affect the training loss value. Moreover, Figure. 4b to Figure. 4i shows that SOS can help stabilize the training process when it is unstable. When the training process without SOS fluctuated a lot, SOS smoothed its loss landscape and

stabilized the highly erratic training process. Moreover, for Figure. 4d, SOS even makes the model converge earlier.

**Generalization ability test.** Next, we demonstrate that SOS can also improve the generalization ability of neural networks when applied in the training process. We apply SGD to train the same four CNN models under the CIFAR10 and CIFAR100 datasets with the same data augmentation strategies. Following (Huang et al., 2017), the weight decay is $10^{-4}$ and a Nesterov momentum of 0.9 without damping. The batch size is set to be 64 and the models are trained for 300 epochs. The initial learning rate is set to be 0.1 and divided by 10 and is divided by 10 at 50% and 75% of the total number of training epochs. All images are applied with a simple random horizontal flip and normalized using their mean and standard deviation. We focus on the comparisons between four different training schemes, namely the standard SGD scheme, SOS on SGD schemes, SAM scheme (Foret et al., 2021), and SOS on SAM schemes. Given that both SGD and SAM search optimal parameters within local regions, our SOS operates globally and thus can be integrated with SGD and SAM to attain better generalization.

Table 2: Testing accuracy of different CNN models on CIFAR10 and CIFAR100 when implementing the four training schemes.

| | | CIFAR100 | | CIFAR10 |
|---|---|---|---|---|
| VGG-16 | Basic | Mixup augmentation | Basic | Mixup augmentation |
| SGD | 70.2 | 72.7 | 91.8 | 93.1 |
| SOS + SGD | 71.5 | **78.3** | 92.1 | 93.6 |
| SAM | 74.8 | 74.3 | 94.4 | 94.8 |
| SOS+SAM | 74.9 | 75.3 | 94.7 | **95.0** |
| ResNet-18 | Basic | Mixup augmentation | Basic | Mixup augmentation |
| SGD | 77.5 | 79.1 | 95.0 | 95.7 |
| SOS + SGD | 78.1 | 79.8 | 95.2 | 96.2 |
| SAM | 78.6 | 80.2 | 95.9 | 96.1 |
| SOS+SAM | 79.2 | **80.4** | 96.1 | **96.4** |
| ResNet-50 | Basic | Mixup augmentation | Basic | Mixup augmentation |
| SGD | 77.9 | 81.3 | 95.1 | 96.0 |
| SOS + SGD | 78.3 | 81.9 | 95.7 | 96.3 |
| SAM | 78.5 | 81.7 | 95.8 | 96.5 |
| SOS+SAM | 78.9 | **82.3** | 96.2 | **96.8** |
| DenseNet-201 | Basic | Mixup augmentation | Basic | Mixup augmentation |
| SGD | 80.1 | 82.0 | 95.3 | 96.2 |
| SOS + SGD | 80.4 | 82.5 | 95.8 | 96.6 |
| SAM | 80.8 | 82.8 | 96.1 | 96.8 |
| SOS+SAM | 82.0 | **83.1** | 96.5 | **97.0** |

As we can see in Table. 2, all the accuracy of the four CNN network architectures on CIFAR10 and CIFAR100 have been improved with SOS compared to the standard SGD and SAM schemes without SOS. The test accuracy is also increased, for example, the test accuracy of VGG-16 on CIFAR100 has been improved from 72.7% to 78.3%

**Parameters' weight study** The model parameters' weight with and without SOS is shown in Figure. 6. The Frobenius norm of the last linear layer's weight increases a lot before dividing the learning rate by 10. It has been slowed down when the learning rate is divided and the weight starts to decrease rapidly for the model with OS. When SOS is applied, the increase rate is slowed down for the DenseNet. For VGG, the weight does not increase but decreases from the first epoch.

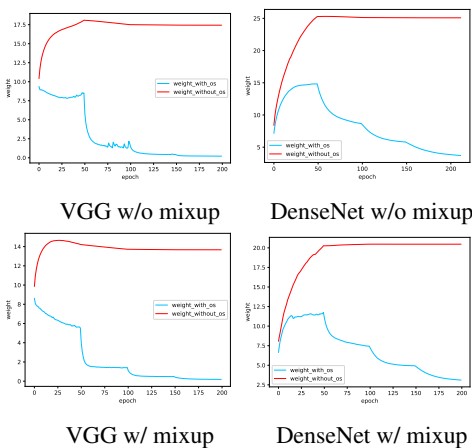

VGG w/o mixup     DenseNet w/o mixup

VGG w/ mixup     DenseNet w/ mixup

Figure 6: Weight visualization.

Table 3: Tiny-Imagenet classification.

|  | ResNet50 | Dense121 | Vit-B |
|---|---|---|---|
| SGD | 52.4 | 67.4 | 53.7 |
| SOS + SGD | 53.8 | 71.1 | 63.1 |
| SAM | 66.8 | 73.7 | 80.7 |
| SOS + SAM | **67.1** | **75.4** | **81.5** |

Table 4: Object detection.

| VOC | Yolo-V5s | Yolo-V5x |
|---|---|---|
| mAP w/ SOS | $83.4_{\pm 0.2}$ | $87.1_{\pm 0.2}$ |
| mAP w/o SOS | $\mathbf{83.7}_{\pm 0.1}$ | $\mathbf{87.4}_{\pm 0.2}$ |

### 5.2.2 TINY-IMAGENET

We evaluate our method on the Tiny-ImageNet classification dataset using three deep vision models: ResNet50 (He et al., 2016), DenseNet121(Huang et al., 2017), and Vision Transformer architecture ViT-B with pre-trained model (Dosovitskiy et al., 2020). For data augmentation, we adopt the Cut-Mix augmentation (Yun et al., 2019). Likewise, we still adopt the four training schemes for comparisons. All the models are trained within 200 epochs with a cosine learning rate schedule.

Table 3 reports the test accuracy on Tiny-Imagenet classification with different models and different training schemes. As we can see from the table, applying SOS in the standard SGD training can improve generalization. For example, the test accuracy of DenseNet121 is improved from 67.4% to 71.1% by simply applying SOS in the standard SGD training process. Moreover, the accuracy can be improved from 73.7% to 75.4% by applying SOS in the SAM training process. Again, this confirms the effectiveness of our scheme for practical application.

### 5.2.3 OBJECT DETECTION

Our method improves generalization performance on other recognition tasks. We do experiments on other computer vision tasks, such as object detection to validate the good generalization performance of our OS method. Table 4 shows the object detection baseline results with and without OS on PASCAL VOC dataset (Everingham et al., 2010). We adopt YOLOv5s and YOLOv5x (Jocher et al., 2020) as the detection model and We see that the performance (mAP) of the two models is improved when applied with SOS.

## 6 CONCLUSION

In this paper, we introduce a novel technique called optimum shifting, to move the parameters of neural networks from sharper minima to flatter minima while keeping the training loss value unchanged. It treats the matrix multiplications in the network as systems of under-determined linear equations and modifies parameters in the solution space. To reduce the computational costs and increase the degrees of freedom for optimum shifting, we proposed stochastic optimum shifting, which selects a small batch for optimum shifting. The neural collapse phenomenon guarantees that the proposed stochastic optimum shifting remains empirical loss unchanged. We perform experiments to show that stochastic optimum shifting improves the generalization ability on different vision tasks, reduces the Hessian trace, keeps the empirical loss unchanged and stabilizes the training process.

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

# A APPENDIX

## A.1 PROOF OF THEOREM 1

We denote $x_{l,i}$ as the output of $l$-th layer of neural network with input data sample $x_i$:

$$x_{l,i} = \sigma(F_l * \cdots \sigma(F_1 * x_i) \cdots). \tag{16}$$

The Hessian trace of loss function $\mathrm{tr}(H_L)$ can be represented as:

$$\mathrm{tr}(H_L) = \sum_{p=1}^{m_{L+1}} \mathrm{tr}(\nabla_{\mathbf{v}_p}^2 L) + \sum_{l_0=1}^{l} \mathrm{tr}(\nabla_{\mathrm{vec}(F_{l_0})}^2 L), \tag{17}$$

where $\mathbf{v}_p$ denotes the $p$-th column of $\mathbf{v}$. The gradient of $L$ with respect to $\mathbf{v}$ is

$$\frac{\partial L}{\partial \mathbf{v}_p} = \frac{1}{n} \sum_{i=1}^{n} \frac{\partial L(f(x_i), y_i)}{\partial f_p(x_i)^T} \frac{\partial f_p(x_i)}{\partial \mathbf{v}_p} = \frac{1}{n} \sum_{i=1}^{n} \frac{\partial L(f(x_i), y_i)}{\partial f_p(x_i)^T} \mathrm{vec}(x_{L,i})^T, \tag{18}$$

where $f_p(x_i)$ denotes the $p$-th element in $f(x_i)$. So, the second-order derivative is as follows:

$$\nabla_{\mathbf{v}_p}^2 L = \frac{1}{n} \sum_{i=1}^{n} \frac{\partial^2 L(f(x_i), y_i)}{\partial f_p(x_i)^T \partial f_p(x_i)^T} \mathrm{vec}(x_{L,i}) \mathrm{vec}(x_{L,i})^T. \tag{19}$$

Because the output of the classifier does not change, $\nabla_{\mathbf{v}}^2 L$ is independent of $\mathbf{v}$ and will not change. For the second part:

$$\nabla_{\mathrm{vec}(F_{l_0})} L = \frac{1}{n} \sum_{i=1}^{n} \frac{\partial L(f(x_i), y_i)}{\partial f(x_i)^T} \left\{ \mathbf{v}^T \prod_{q=l+1}^{L-1} \left[ (\Sigma_{q,i} \mathrm{vec}(F_q)^T \otimes I_q) M_q \right] \left[ \Sigma_{l,i} I_l \otimes \phi(x_{l,i}) \right] \right\}, \tag{20}$$

where $\Sigma_{l,i} = \mathrm{Diag}\left[ \mathbb{1}\left\{ \mathrm{vec}(x_{l,i}) > 0 \right\} \right]$, $I_l \in \mathbb{R}^{|x_{l+1,i}| \times |x_{l+1,i}|}$[1] is an identity matrix, and $M_l$ is an indicator matrix satisfying $\mathrm{vec}(\phi(x_{l,i})) = M_l \mathrm{vec}(x_{l,i})$ (Wu, 2020; Vedaldi & Lenc, 2015). So the second order derivative is:

$$\nabla_{\mathrm{vec}(F_{l_0})}^2 L = \frac{1}{n} \sum_{i=1}^{n} \frac{\partial^2 L(f(x_i), y_i)}{\partial f(x_i)^T \partial f(x_i)^T} \left\{ \mathbf{v}^T \prod_{q=l+1}^{L-1} \left[ (\Sigma_{q,i} \mathrm{vec}(F_q)^T \otimes I_q) M_q \right] \left[ \Sigma_{l,i} I_l \otimes \phi(x_{l,i}) \right] \right\}^T \cdot$$
$$\left\{ \mathbf{v}^T \prod_{q=l+1}^{L-1} \left[ (\Sigma_{q,i} \mathrm{vec}(F_q)^T \otimes I_q) M_q \right] \left[ \Sigma_{l,i} I_l \otimes \phi(x_{l,i}) \right] \right\}. \tag{21}$$

We use $\mathcal{S}$ to denote $\prod_{q=l+1}^{L-1} \left[ (\Sigma_{q,i} \mathrm{vec}(F_q)^T \otimes I_q) M_q \right] \left[ \Sigma_{l,i} I_l \otimes \phi(x_{l,i}) \right]$. And $\mathrm{tr}(H_L)$ can be expressed as:

$$\mathrm{tr}(H_L) = \frac{1}{n} \sum_{i=1}^{n} \sum_{p=1}^{m_{L+1}} \frac{\partial^2 L(f(x_i), y_i)}{\partial f_p(x_i)^T \partial f_p(x_i)^T} \|\mathrm{vec}(x_{L,i})\|^2 + \frac{1}{n} \sum_{l_0=1}^{l} \sum_{i=1}^{n} \mathrm{tr}\left\{ \frac{\partial^2 L(f(x_i), y_i)}{\partial f(x_i)^T \partial f(x_i)^T} \mathcal{S}^T \mathbf{v} \mathbf{v}^T \mathcal{S} \right\} \tag{22}$$

So the trace of Hessian can be upper bounded by:

$$\mathrm{tr}(H_L) \leq \frac{1}{n} \sum_{i=1}^{n} \sum_{p=1}^{m_{L+1}} \frac{\partial^2 L(f(x_i), y_i)}{\partial f_p(x_i)^T \partial f_p(x_i)^T} \|\mathrm{vec}(x_{L,i})\|^2 + \frac{\|\mathbf{v}\|^2}{n} \sum_{l_0=1}^{l} \sum_{i=1}^{n} \mathrm{tr}\left\{ \frac{\partial^2 L(f(x_i), y_i)}{\partial f(x_i)^T \partial f(x_i)^T} \mathcal{S}^T \mathcal{S} \right\}, \tag{23}$$

Thus, we have finished the proof of the upper bound. For the lower bound, because $\mathcal{S}\mathcal{S}^T$ is semi-positive definite, so we factorized it as:

$$\mathcal{S}\mathcal{S}^T = Q \begin{bmatrix} \Lambda & 0 \\ 0 & 0 \end{bmatrix} Q^T \tag{24}$$

---

[1] $|A|$ denotes the number of elements in $A$.

where $Q$ is orthogonal matrix and $\Lambda$ is is a diagonal matrix. The second part of Eq. (22) can be lower bounded as:

$$\nabla^2_{\text{vec}(\boldsymbol{F}_l)} L = \frac{1}{n} \sum_{l_0=1}^{l} \sum_{i=1}^{n} \frac{\partial^2 L(f(\boldsymbol{x}_i), \boldsymbol{y}_i)}{\partial f(\boldsymbol{x}_i)^T \partial f(\boldsymbol{x}_i)^T} \text{tr}(\mathcal{S}^T \mathbf{v}\mathbf{v}^T \mathcal{S}) \tag{25}$$

$$= \frac{1}{n} \sum_{l_0=1}^{l} \sum_{i=1}^{n} \frac{\partial^2 L(f(\boldsymbol{x}_i), \boldsymbol{y}_i)}{\partial f(\boldsymbol{x}_i)^T \partial f(\boldsymbol{x}_i)^T} \text{tr}(\mathbf{v}\mathbf{v}^T Q \begin{bmatrix} \Lambda & 0 \\ 0 & 0 \end{bmatrix} Q^T) \tag{26}$$

$$= \frac{1}{n} \sum_{l_0=1}^{l} \sum_{i=1}^{n} \frac{\partial^2 L(f(\boldsymbol{x}_i), \boldsymbol{y}_i)}{\partial f(\boldsymbol{x}_i)^T \partial f(\boldsymbol{x}_i)^T} \frac{\text{tr}(\mathbf{v}\mathbf{v}^T Q \begin{bmatrix} \Lambda & 0 \\ 0 & 0 \end{bmatrix} Q^T) \text{tr}(\begin{bmatrix} \Lambda^{-1} & 0 \\ 0 & 0 \end{bmatrix})}{\text{tr}(\begin{bmatrix} \Lambda^{-1} & 0 \\ 0 & 0 \end{bmatrix})} \tag{27}$$

$$\geq \frac{1}{n} \sum_{l_0=1}^{l} \sum_{i=1}^{n} \frac{\partial^2 L(f(\boldsymbol{x}_i), \boldsymbol{y}_i)}{\partial f(\boldsymbol{x}_i)^T \partial f(\boldsymbol{x}_i)^T} \frac{\|\mathbf{v}\|^2}{\text{tr}(\Lambda^{-1})} \tag{28}$$

$$\tag{29}$$

Thus, we have finished the proof of lower bound.

The first part is independent with $\|\mathbf{v}\|^2$. And the second part is linearly dependent on $\|\mathbf{v}\|^2$. So if $\|\mathbf{v}\|^2$ is minimized, then the upper and lower bound of the Hessian trace will also be minimized, and we end the proof.

### A.2 PROOF OF THEOREM 2

For ResNet, which uses skip connection that bypasses the non-linear transformations:

$$\boldsymbol{x}_{2k+1,i} = \sigma(\boldsymbol{F}_{2k} * \sigma(\boldsymbol{F}_{2k-1} * \boldsymbol{x}_{2k-1,i}) + \boldsymbol{x}_{2k-1,i}) \tag{30}$$

$$= \sigma(\boldsymbol{F}_{2k} * \boldsymbol{x}_{2k,i} + \boldsymbol{x}_{2k-1,i}) \tag{31}$$

It can also be represented as:

$$\text{vec}(\boldsymbol{x}_{2k+1,i}) = \Sigma_{2k,i} \left[ \text{vec}(\boldsymbol{F}_{2k} * \boldsymbol{x}_{2k,i}) + \text{vec}(\boldsymbol{x}_{2k-1,i}) \right] \tag{32}$$

$$= \begin{cases} \Sigma_{2k,i} I_{2k} \otimes \phi(\boldsymbol{x}_{2k,i}) \text{vec}(\boldsymbol{F}_{2k}) + \Sigma_{2k,i} \text{vec}(\boldsymbol{x}_{2k-1,i}) \\ \Sigma_{2k,i} (\boldsymbol{F}_{2k}^T \otimes I_{2k}) M_{2k} \text{vec}(\boldsymbol{x}_{2k,i}) + \Sigma_{2k,i} \text{vec}(\boldsymbol{x}_{2k-1,i}) \end{cases} \tag{33}$$

So the gradient with respect to $\text{vec}(\boldsymbol{F})$ and $\text{vec}(\boldsymbol{x}_{2k-1})$ is:

$$\frac{\partial \text{vec}(\boldsymbol{x}_{2k+1,i})}{\partial \text{vec}(\boldsymbol{x}_{2k-1,i})^T} = \Sigma_{2k,i}(\boldsymbol{F}_{2k}^T \otimes I_{2k}) M_{2k} \Sigma_{2k-1,i}(\boldsymbol{F}_{2k-1}^T \otimes I_{2k-1}) M_{2k-1} + \Sigma_{2k,i} \tag{34}$$

$$\frac{\partial \text{vec}(\boldsymbol{x}_{2k+1,i})}{\partial \text{vec}(\boldsymbol{F}_{2k})^T} = \Sigma_{2k,i} I_{2k} \otimes \phi(\boldsymbol{x}_{2k,i}) \tag{35}$$

$$\frac{\partial \text{vec}(\boldsymbol{x}_{2k+1,i})}{\partial \text{vec}(\boldsymbol{F}_{2k-1})^T} = \Sigma_{2k,i}(\boldsymbol{F}_{2k}^T \otimes I_{2k}) M_{2k} \Sigma_{2k-1,i} I_{2k-1} \otimes \phi(\boldsymbol{x}_{2k-1,i}) \tag{36}$$

We assume that $l$ is multiples of 2, i.e. $\exists t$ s.t. $l = 2t$, and we represent the neural network as follows:

$$f(\boldsymbol{x}_i) = \mathbf{v}^T \text{vec}(\sigma(\boldsymbol{F}_{2t} * \boldsymbol{x}_{2t,i} + \boldsymbol{x}_{2t-1,i})) \tag{37}$$

The gradient of $f$ with respect to $\text{vec}(\boldsymbol{F}_{2l_0})$ and $\text{vec}(\boldsymbol{F}_{2l_0-1})$ are:

$$\nabla_{\text{vec}(\boldsymbol{F}_{2l_0})} f = \frac{\partial f}{\partial \text{vec}(\boldsymbol{x}_{2t+1,i})^T} \left[ \prod_{r=l+1}^{t} \left( \frac{\partial \text{vec}(\boldsymbol{x}_{2r+1,i})}{\partial \text{vec}(\boldsymbol{x}_{2r-1,i})^T} \right) \right] \frac{\partial \text{vec}(\boldsymbol{x}_{2l_0+1,i})}{\partial \text{vec}(\boldsymbol{F}_{2l})^T} \tag{38}$$

$$\nabla_{\text{vec}(\boldsymbol{F}_{2l_0-1})} f = \frac{\partial f}{\partial \text{vec}(\boldsymbol{x}_{2t+1,i})^T} \left[ \prod_{r=l+1}^{t} \left( \frac{\partial \text{vec}(\boldsymbol{x}_{2r+1,i})}{\partial \text{vec}(\boldsymbol{x}_{2r-1,i})^T} \right) \right] \frac{\partial \text{vec}(\boldsymbol{x}_{2l_0+1,i})}{\partial \text{vec}(\boldsymbol{F}_{2l-1})^T} \tag{39}$$

$$\tag{40}$$

We denote $\left[\prod_{r=l+1}^{L-1}\left(\frac{\partial \boldsymbol{x}_{2r+1,i}}{\partial \boldsymbol{x}_{2r-1,i}^T}\right)\right]\left[\mathbb{1}_{r\%2=0}\left[\frac{\partial \mathrm{vec}(\boldsymbol{x}_{2l_0+1,i})}{\partial \mathrm{vec}(\boldsymbol{F}_{2l})^T}\right] + \mathbb{1}_{r\%2=1}\left[\frac{\partial \mathrm{vec}(\boldsymbol{x}_{2l_0+1,i})}{\partial \mathrm{vec}(\boldsymbol{F}_{2l-1})^T}\right]\right]$ as $\mathcal{S}$. The Hessian trace can be represented as

$$\mathrm{tr}(H_L) = \frac{1}{n}\sum_{q=1}^{m_{L+1}}\sum_{i=1}^{n}\frac{\partial^2 L(f(\boldsymbol{x}_i), \boldsymbol{y}_i)}{\partial f_q(\boldsymbol{x}_i)^T \partial f_q(\boldsymbol{x}_i)^T}\|\mathrm{vec}(\boldsymbol{x}_{L,i})\|^2 + \tag{41}$$

$$\frac{1}{n}\sum_{i=1}^{n}\sum_{l_0=1}^{l}\mathrm{tr}\left\{\frac{\partial^2 L(f(\boldsymbol{x}_i), \boldsymbol{y}_i)}{\partial f(\boldsymbol{x}_i)^T \partial f(\boldsymbol{x}_i)^T}\mathcal{S}^T \mathbf{v}\mathbf{v}^T S\right\} \tag{42}$$

This has the same form as Eq. (22). So the Hessian trace is upper bounded by:

$$\mathrm{tr}(\boldsymbol{H}_L) \leq \frac{1}{n}\sum_{i=1}^{n}\sum_{p=1}^{m_{L+1}}\frac{\partial^2 L(f(\boldsymbol{x}_i), \boldsymbol{y}_i)}{\partial f_p(\boldsymbol{x}_i)^T \partial f_p(\boldsymbol{x}_i)^T}\|\mathrm{vec}(\boldsymbol{x}_{L,i})\|^2 + \frac{\|\mathbf{v}\|^2}{n}\sum_{l_0=1}^{l}\sum_{i=1}^{n}\mathrm{tr}\left\{\frac{\partial^2 L(f(\boldsymbol{x}_i), \boldsymbol{y}_i)}{\partial f(\boldsymbol{x}_i)^T \partial f(\boldsymbol{x}_i)^T}\mathcal{S}^T \mathcal{S}\right\}, \tag{43}$$

and lower bounded by:

$$\nabla^2_{\mathrm{vec}(\boldsymbol{F}_l)}L \geq \frac{1}{n}\sum_{i=1}^{n}\sum_{p=1}^{m_{L+1}}\frac{\partial^2 L(f(\boldsymbol{x}_i), \boldsymbol{y}_i)}{\partial f_p(\boldsymbol{x}_i)^T \partial f_p(\boldsymbol{x}_i)^T}\|\mathrm{vec}(\boldsymbol{x}_{L,i})\|^2 + \frac{1}{n}\sum_{l_0=1}^{l}\sum_{i=1}^{n}\frac{\partial^2 L(f(\boldsymbol{x}_i), \boldsymbol{y}_i)}{\partial f(\boldsymbol{x}_i)^T \partial f(\boldsymbol{x}_i)^T}\frac{\|\mathbf{v}\|^2}{\mathrm{tr}(\Lambda^{-1})} \tag{44}$$

### A.3 PROOF OF THEOREM 3

When neural networks using dense connective pattern, which is represented as follows:

$$\boldsymbol{x}_{l+1,i} = \sigma(\boldsymbol{F}_l * [\boldsymbol{x}_{l,i}, \cdots, \boldsymbol{x}_{1,i}]) \tag{45}$$

$$= \sigma(\sum_{j=1}^{l}\boldsymbol{F}_l * \boldsymbol{x}_{j,i}) \tag{46}$$

We vectorize it and represented the dense connection as:

$$\mathrm{vec}(\boldsymbol{x}_{l+1,i}) = \Sigma_{l,i}\left[\sum_{j=1}^{l}(\boldsymbol{F}_j^T \otimes I_j)\mathrm{vec}(\phi(\boldsymbol{x}_{j,i}))\right] \tag{47}$$

$$= \sum_{j=1}^{l}\left[\Sigma_{l,i}(\boldsymbol{F}_j^T \otimes I_j)\mathrm{vec}(\phi(\boldsymbol{x}_{j,i}))\right] \tag{48}$$

The neural network with $L$ convolutional layer can be represented as:

$$f(\boldsymbol{x}_i) = \mathbf{v}^T \mathrm{vec}(\boldsymbol{x}_{L,i}) \tag{49}$$

The gradient of $p$-th layer output with respect to $q$-th layer output is:

$$\frac{\partial \mathrm{vec}(\boldsymbol{x}_{p,i})}{\partial \mathrm{vec}(\boldsymbol{x}_{q,i})^T} = \sum_{o=q}^{p}\prod_{j=q}^{o}\Sigma_{j,i}(\boldsymbol{F}_j^T \otimes I_j)M_j \tag{50}$$

So the gradient of $f$ with respect to $\mathrm{vec}(\boldsymbol{F}_l)$ is:

$$\frac{\partial f(\boldsymbol{x}_i)}{\partial \mathrm{vec}(\boldsymbol{F}_l)^T} = \frac{\partial f(\boldsymbol{x}_i)}{\partial \mathrm{vec}(\boldsymbol{x}_{L+1})^T}\frac{\partial \mathrm{vec}(\boldsymbol{x}_{L+1})}{\partial \mathrm{vec}(\boldsymbol{x}_{l+1})^T}\frac{\partial \mathrm{vec}(\boldsymbol{x}_{l+1})}{\partial \mathrm{vec}(\boldsymbol{F}_l)^T} \tag{51}$$

$$= \mathbf{v}^T\left[\sum_{o=l+1}^{L}\prod_{j=q}^{o}\Sigma_{j,i}(\boldsymbol{F}_j^T \otimes I_j)M_j\right]\Sigma_{l,i}I_l \otimes \phi(\boldsymbol{x}_{l,i}) \tag{52}$$

$$:= \mathbf{v}^T \mathcal{S} \tag{53}$$

The trace of Hessian can be represented as

$$\text{tr}(H_L) = \frac{1}{n} \sum_{q=1}^{m_{L+1}} \sum_{i=1}^{n} \frac{\partial^2 L(f(\boldsymbol{x}_i), \boldsymbol{y}_i)}{\partial f_q(\boldsymbol{x}_i)^T \partial f_q(\boldsymbol{x}_i)^T} \|\text{vec}(\boldsymbol{x}_{L,i})\|^2 + \tag{54}$$

$$\frac{1}{n} \sum_{i=1}^{n} \sum_{l_0=1}^{l} \text{tr} \left\{ \frac{\partial^2 L(f(\boldsymbol{x}_i), \boldsymbol{y}_i)}{\partial f(\boldsymbol{x}_i)^T \partial f(\boldsymbol{x}_i)^T} \mathcal{S}^T \mathbf{v} \mathbf{v}^T \mathcal{S} \right\} \tag{55}$$

This has the same form as Eq. (22). So the Hessian trace is upper bounded by:

$$\text{tr}(\boldsymbol{H}_L) \leq \frac{1}{n} \sum_{i=1}^{n} \sum_{p=1}^{m_{L+1}} \frac{\partial^2 L(f(\boldsymbol{x}_i), \boldsymbol{y}_i)}{\partial f_p(\boldsymbol{x}_i)^T \partial f_p(\boldsymbol{x}_i)^T} \|\text{vec}(\boldsymbol{x}_{L,i})\|^2 + \frac{\|\mathbf{v}\|^2}{n} \sum_{l_0=1}^{l} \sum_{i=1}^{n} \text{tr} \left\{ \frac{\partial^2 L(f(\boldsymbol{x}_i), \boldsymbol{y}_i)}{\partial f(\boldsymbol{x}_i)^T \partial f(\boldsymbol{x}_i)^T} \mathcal{S}^T \mathcal{S} \right\}, \tag{56}$$

and lower bounded by:

$$\nabla^2_{\text{vec}(\boldsymbol{F}_l)} L \geq \frac{1}{n} \sum_{i=1}^{n} \sum_{p=1}^{m_{L+1}} \frac{\partial^2 L(f(\boldsymbol{x}_i), \boldsymbol{y}_i)}{\partial f_p(\boldsymbol{x}_i)^T \partial f_p(\boldsymbol{x}_i)^T} \|\text{vec}(\boldsymbol{x}_{L,i})\|^2 + \frac{1}{n} \sum_{l_0=1}^{l} \sum_{i=1}^{n} \frac{\partial^2 L(f(\boldsymbol{x}_i), \boldsymbol{y}_i)}{\partial f(\boldsymbol{x}_i)^T \partial f(\boldsymbol{x}_i)^T} \frac{\|\mathbf{v}\|^2}{\text{tr}(\Lambda^{-1})} \tag{57}$$

### A.4 PROOF OF THEOREM 4

For a fully connected neural network. Let $\{(\mathbf{x}_1, \boldsymbol{y}_1), \cdots, (\mathbf{x}_n, \boldsymbol{y}_n)\}$ be a set of n training samples. We consider l-hidden-layer neural networks as follows.

$$f(\boldsymbol{x}) = \mathbf{v}^T \sigma(\boldsymbol{W}_l^T \sigma(\boldsymbol{W}_{l-1}^T \cdots \sigma(\boldsymbol{W}_1^T \boldsymbol{x}) \cdots)) \tag{58}$$

where $\sigma(x) = \max\{0, x\}$ is the entry-wise ReLU activation. $\boldsymbol{W}_l \in \mathbb{R}^{m_{l-1} \times m_l}$. Given input $\boldsymbol{x}_i$, We denote the output after the $l_0$-th layer using $\boldsymbol{x}_{l_0,i}$

$$\boldsymbol{x}_{l_0,i} = \sigma(\boldsymbol{W}_{l_0}^T \sigma(\boldsymbol{W}_{l_0-1}^T \cdots \sigma(\boldsymbol{W}_1^T \boldsymbol{x}_i) \cdots)) \tag{59}$$

$$= (\prod_{r=1}^{l} \Sigma_{r,i} \boldsymbol{W}_r^T) \boldsymbol{x}_i \tag{60}$$

where $\Sigma_{1,i} = \text{Diag}(\mathbb{1}\{\boldsymbol{W}_1^T \boldsymbol{x}_i > 0\})$, and $\Sigma_{l_0,i} = \text{Diag}\left[\mathbb{1}\left\{\boldsymbol{W}_{l_0}^T (\prod_{r=1}^{l_0-1} \Sigma_{i,r} \boldsymbol{W}_r^T) \boldsymbol{x}_i > 0\right\}\right]$. We have $f(\boldsymbol{x}_i) = \mathbf{v}^T \boldsymbol{x}_{l,i}$. The Hessian trace of loss function $\text{tr}(\boldsymbol{H}_L)$ can be represented as:

$$\text{tr}(\boldsymbol{H}_L) = \sum_{p=1}^{m_{l+1}} \text{tr}(\nabla^2_{\mathbf{v}_p} L) + \sum_{l_0=1}^{l} \text{tr}(\nabla^2_{\text{vec}(\boldsymbol{F}_{l_0})} L), \tag{61}$$

where $\mathbf{v}_p$ denotes the $p$-th column of $\mathbf{v}$. The gradient of $L$ with respect to $\mathbf{v}$ is

$$\frac{\partial L}{\partial \mathbf{v}_p} = \frac{1}{n} \sum_{i=1}^{n} \frac{\partial L(f(\boldsymbol{x}_i), \boldsymbol{y}_i)}{\partial f_p(\boldsymbol{x}_i)^T} \frac{\partial f_p(\boldsymbol{x}_i)}{\partial \mathbf{v}_p} = \frac{1}{n} \sum_{i=1}^{n} \frac{\partial L(f(\boldsymbol{x}_i), \boldsymbol{y}_i)}{\partial f_p(\boldsymbol{x}_i)^T} \text{vec}(\boldsymbol{x}_{L,i})^T, \tag{62}$$

where $f_p(\boldsymbol{x}_i)$ denotes the $p$-th element in $f(\boldsymbol{x}_i)$. So, the second-order derivative is as follows:

$$\nabla^2_{\mathbf{v}_p} L = \frac{1}{n} \sum_{i=1}^{n} \frac{\partial^2 L(f(\boldsymbol{x}_i), \boldsymbol{y}_i)}{\partial f_p(\boldsymbol{x}_i)^T \partial f_p(\boldsymbol{x}_i)^T} \text{vec}(\boldsymbol{x}_{L,i}) \text{vec}(\boldsymbol{x}_{L,i})^T. \tag{63}$$

The gradient of $L$ with respect to the $p$-th column of $\boldsymbol{W}_l$ (denoted by $\boldsymbol{W}_{p,l}$) is :

$$\nabla_{\boldsymbol{W}_{p,l}} L = \frac{1}{n} \sum_{i=1}^{n} \frac{\partial L(f(\boldsymbol{x}_i), \boldsymbol{y}_i)}{\partial f_p(\boldsymbol{x}_i)^T} \left[ \mathbf{v}^T (\prod_{r=l+1}^{L} \Sigma_{r,i} \boldsymbol{W}_r^T) \Sigma_{l,i} \mathbf{0}_p(\boldsymbol{x}_{l,i}) \right] \tag{64}$$

where $\mathbf{0}_p(\boldsymbol{x}_{l,i})$ denotes a $m_{l+1} \times m_l$ matrix in which the $p$-th row equals to the $p$-th row of $\boldsymbol{x}_{l,i}$ and the other $m_{l+1} - 1$ rows equals are all zeros. And the Hessian of $L$ with respect to $\boldsymbol{W}_{p,l}$ is

$$\nabla^2_{\boldsymbol{W}_{p,l}} L = \frac{1}{n} \sum_{i=1}^{n} \frac{\partial^2 L(f(\boldsymbol{x}_i), \boldsymbol{y}_i)}{\partial f(\boldsymbol{x}_i)^T \partial f(\boldsymbol{x}_i)^T} \left[ \mathbf{v}^T (\prod_{r=l+1}^{L} \Sigma_{r,i} \boldsymbol{W}_r^T) \Sigma_{l,i} \mathbf{0}_p(\boldsymbol{x}_{l,i}) \right]^T \left[ \mathbf{v}^T (\prod_{r=l+1}^{L} \Sigma_{r,i} \boldsymbol{W}_r^T) \Sigma_{l,i} \mathbf{0}_p(\boldsymbol{x}_{l,i}) \right] \tag{65}$$

$$:= \frac{1}{n} \sum_{i=1}^{n} \frac{\partial^2 L(f(\boldsymbol{x}_i), \boldsymbol{y}_i)}{\partial f(\boldsymbol{x}_i)^T \partial f(\boldsymbol{x}_i)^T} \mathcal{S}^T \mathbf{v} \mathbf{v}^T \mathcal{S} \tag{66}$$

Thus, the trace of Hessian has the same form as Eq. (22). So the it is upper bounded by:

$$\text{tr}(\boldsymbol{H}_L) \leq \frac{1}{n} \sum_{i=1}^{n} \sum_{p=1}^{m_{L+1}} \frac{\partial^2 L(f(\boldsymbol{x}_i), \boldsymbol{y}_i)}{\partial f_p(\boldsymbol{x}_i)^T \partial f_p(\boldsymbol{x}_i)^T} \|\text{vec}(\boldsymbol{x}_{L,i})\|^2 + \frac{\|\mathbf{v}\|^2}{n} \sum_{l_0=1}^{l} \sum_{i=1}^{n} \text{tr} \left\{ \frac{\partial^2 L(f(\boldsymbol{x}_i), \boldsymbol{y}_i)}{\partial f(\boldsymbol{x}_i)^T \partial f(\boldsymbol{x}_i)^T} \mathcal{S}^T \mathcal{S} \right\}, \tag{67}$$

and lower bounded by:

$$\nabla^2_{\text{vec}(\boldsymbol{F}_l)} L \geq \frac{1}{n} \sum_{i=1}^{n} \sum_{p=1}^{m_{L+1}} \frac{\partial^2 L(f(\boldsymbol{x}_i), \boldsymbol{y}_i)}{\partial f_p(\boldsymbol{x}_i)^T \partial f_p(\boldsymbol{x}_i)^T} \|\text{vec}(\boldsymbol{x}_{L,i})\|^2 + \frac{1}{n} \sum_{l_0=1}^{l} \sum_{i=1}^{n} \frac{\partial^2 L(f(\boldsymbol{x}_i), \boldsymbol{y}_i)}{\partial f(\boldsymbol{x}_i)^T \partial f(\boldsymbol{x}_i)^T} \frac{\|\mathbf{v}\|^2}{\text{tr}(\Lambda^{-1})} \tag{68}$$

# B  HESSIAN ANALYSIS OF SOS DURING TRAINING

The evolution of the trace of Hessian during training with and without SOS is show in Fig. 7. We train VGG16 and ResNet18 on CIFAR10 and CIFAR100 dataset for 200 epochs and visualize the trace of Hessian during training. It shows that the trace of Hessian during training is keep increasing. However, when we apply SOS to the model during training, the trace of Hessian has been minimized, which indicates a better generalization ability.

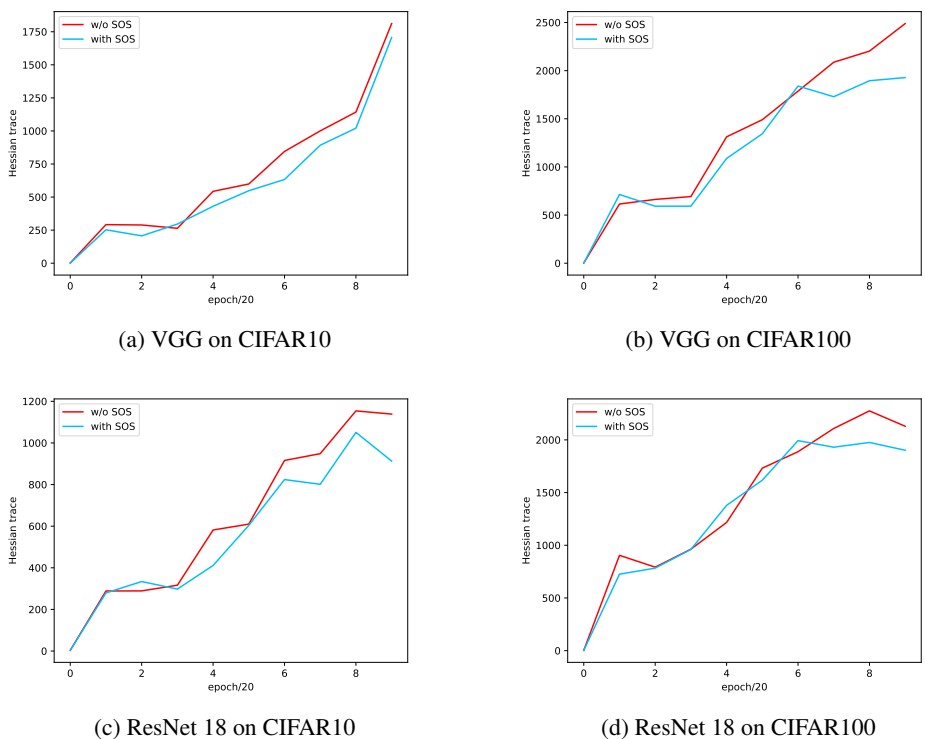

(a) VGG on CIFAR10

(b) VGG on CIFAR100

(c) ResNet 18 on CIFAR10

(d) ResNet 18 on CIFAR100

Figure 7: Visualization of the Hessian trace for different models and dataset with and without SOS

## C  SOCIAL IMPACTS

As a new method to change the parameters of neural network and increase the generalization ability, it may be beneficial to models applied in different fields such as computer vision. Besides, it may boost the studies in generalization and loss landscape analysis. Thus, we believe that our algorithm will bring positive impacts on both academia and industry.

