# OpenReview forum: "Optimum Shifting to Stabilize Training and Improve Generalization of Deep Neural Networks"
_ICLR.cc/2024/Conference — Submitted to ICLR 2024_

### Official Review · Reviewer_WtMt · 2023-10-27

**Soundness:** 3 good
**Presentation:** 3 good
**Contribution:** 2 fair
**Rating:** 5
**Confidence:** 4

**Summary:**

This paper proposes a method called optimum shifting (OS) to modify the parameters of a neural network from one point to a flatter one while keeping the training loss unchanged. It minimizes the Frobenius norm of the weight in the final fully connected layer of a neural network without changing the output of the final layer for given training data. Further, a stochastic version (SOS) is proposed to reduce the computational costs and provide more degrees of freedom for optimum shifting. Experiments show the proposed method improves the generalization ability on different vision tasks.

**Strengths:**

1. This paper has a good motivation.
2. SOS is efficient and compatible with traditional regularization techniques.

**Weaknesses:**

This paper considers to modify the parameters of a neural network from one point to a flatter one while keeping the training loss unchanged for some training data. However, the specific implementation of this idea in this paper is very limited. Therefore, I think the current manuscript is not ready for publication.

1. This paper is not very clear due to the inconsistent meanings of symbols and the incomplete description of experimental settings.
2. The idea is not good enough. The implementation of the concept of optimum shifting is very limited.

**Questions:**

1. Theorem results in section 3.2 are very monotonous. Four theorems say almost the same things.
2. The role of gaussian elimination used in SOS is not clear. In fact, I cannot understand why the rows in input matrix A need to be independent.
3. Hessian Analysis for section 5.2 is missing.

---

> ### Author Response · Authors · 2023-11-14
>
> We appreciate the constructive comments on our paper. The raised concerns are addressed as follows.
> >**W1. This paper is not very clear due to the inconsistent meanings of symbols and the incomplete description of experimental settings.**
>
> Thanks for your suggestion, we have detailed the description of experimental settings in line 4-10 in page 8 of our main paper. We demonstrate that SOS can also improve the generalization ability of neural networks when applied in the training process. We apply SGD to train the same four CNN models under the CIFAR10 and CIFAR100 datasets with the same data augmentation strategies. Following Huang et al. [4], the weight decay is 10e-4 and a Nesterov momentum of 0.9 without
> damping. The batch size is set to be 64 and the models are trained for 300 epochs. The initial learning
> rate is set to be 0.1 and divided by 10 and is divided by 10 at 50$\%$ and 75$\%$ of the total number of
> training epochs. All images are applied with a simple random horizontal flip and normalized using
> their mean and standard deviation.
>
> >**W2. The idea is not good enough. The implementation of the concept of optimum shifting is very limited.**
>
> Current sharpness based minimization work mainly focus on image classification tasks such as [1, 2, 3]. Compared with them, we not only perform SOS on image classification tasks but also on object detection tasks to validate the effectiveness of our method. Moreover, our experiment and implementation is also acknowledged by Reviewer KEY9 in Strength 4 : the extensive experimental results on different datasets and network architecture provide a comprehensive demonstration of the proposed method.
>
> >**Q1. Theorem results in section 3.2 are very monotonous. Four theorems say almost the same things.**
>
> We use different model architectures (CNNs, ResNets, DenseNets, et al.) in our experiment to test the effectiveness of SOS, so we have proved the theorems for different neural network models. We will put some relevant theorems in the Appendix in the final version.
>
> >**Q2. The role of gaussian elimination used in SOS is not clear. In fact, I cannot understand why the rows in input matrix A need to be independent.**
>
> When we calculate $A^T(AA^T)^{-1}b$ directly, if each row of $A$ is not independent, $AA^T$ will be irreversible.
> In practice, $A$ may be row-dependent. However, after Gaussian elimination, each non-zero row of input matrix $A$ will be independent and those zero rows will be discarded.  Thus the $A_*$ (the matrix after Gaussian elimination) in our algorithm 1 will be row-independent and is easy to calculate the inverse matrix $(A_*A_*^T)^{-1}$.
>
> >**Q3. Hessian Analysis for section 5.2 is missing.**
>
> Thanks for your suggestion, we have added the Hessian analysis of section 5.2 in the Appendix B. The trace of Hessian during training is keep increasing. However, when we apply SOS to the model during training, the trace of Hessian has been minimized, which indicates a better generalization ability. We thank your good suggestions again.
>
> **References:**
> * [1] Pierre Foret, Ariel Kleiner, Hossein Mobahi, and Behnam Neyshabur. Sharpness-aware minimization for efficiently improving generalization. In ICLR, 2021
> * [2] Jungmin Kwon, Jeongseop Kim, Hyunseo Park, and In Kwon Choi. Asam: Adaptive sharpness aware minimization for scale-invariant learning of deep neural networks. In ICML, 2021.
> * [3] Yang Zhao, Hao Zhang, and Xiuyuan Hu. Penalizing gradient norm for efficiently improving gen-
> eralization in deep learning. In ICML, 2022.
> * [4] Gao Huang, Zhuang Liu, Laurens Van Der Maaten, and Kilian Q Weinberger. Densely connected
> convolutional networks. In CVPR, 2017.

---

> ### Author Response · Authors · 2023-11-21
> **Kind Reminder**
>
> Dear Reviewer,
>
> Thank you again for your great efforts and valuable comments in reviewing our paper. We have carefully addressed the main concerns in detail in the rebuttal and we hope you might find the response satisfactory. As the discussion phase is about to close, we are very much looking forward to hearing from you about any further feedback. We will be very happy to clarify further concerns (if any).
>
> Best,
>
> Authors

---

### Official Review · Reviewer_KEY9 · 2023-10-30

**Soundness:** 3 good
**Presentation:** 3 good
**Contribution:** 3 good
**Rating:** 8
**Confidence:** 5

**Summary:**

This paper proposes a novel technique to change a trained neural network's parameters to a flat loss-landscape region without changing the training loss of the network, which is termed optimal shifting. The main idea is to treat the parameter of a neural network as a linear system and then minimize the trace of its Hessian. To overcome the computational challenge, i.e., the complete estimation of the Hessian matrix required for the whole training dataset, the author proposed an online optimization method motivated by the Stochastic Gradient Descent, where the author use a small batch of data to calculate the optimal shifting. Extensive experiments are conducted on different network architecture and computer vision tasks to demonstrate the effectiveness of the proposed method in improving the generalization of neural networks.

**Strengths:**

1. Overall, the paper is very well written, with a clear demonstration of the main idea, and flows well in presenting the technical details.
2. The perspective of viewing the parameters of a neural network as a linear system is refreshing, though it is not a ground-breaking idea. The proposed method is sound by connecting the flatness with the Hessian and also by providing the theoretical justification for minimizing the Frobenius norm of the weight in the final linear layer that can minimize the trace of the Hessian.
3. The computational challenge is valid and significant, and it is good to see the author provide a very clean summary of the technical challenges for the reader to appreciate the contribution of the present work. The proposed Stochastic Optimum Shifting (SOS) seems very natural to deal with the issue, and it is interesting to see the author bridge the connection between Neural Collapse theory to provide an intuitive justification for the SOS.
4. The extensive experimental results on different datasets and network architecture provide a comprehensive demonstration of the proposed method.

**Weaknesses:**

1. In the introduction, the author claimed that:

"Assume that the rows in $A$ are independent without loss of generality, the equation $Ax=b$ is under-determined if $m<n$ and has infinite solutions for $v$."

The author should provide a more explicit explanation about how we can formally guarantee independence when $A$ is a neural network since, at least it is not very intuitive to the reviewer.

2. The author considers the neural network a linear system, which is fine for the linear layers in the neural network when we omit the non-linear activation function. However, as modeled in Equation (3), the neural network is a stack of linear layers with corresponding nonlinear activation functions. How will the exact choice of the nonlinear activation function have an impact on the theoretical results?

3. From Tables 1 and 2 we can observe that the SOS performs the best when it integrates with the Mixup. Can the author provide any interpretation and reasoning about this phenomenon? Why the basic training strategy can not bring sufficient benefit for the proposed method?

4. The improvement in the object detection task in Table 4 is too marginal. It is better for the author to include the mean and std for this table and also for other tables.

**Questions:**

Please refer to the Weaknesses section for more details.

---

> ### Author Response · Authors · 2023-11-14
>
> We appreciate the positive rating and constructive comments on our paper. The raised concerns are addressed as follows.
>
> >**Q1. In the introduction, the author claimed that:
> "Assume that the rows in $A$ are independent without loss of generality, the equation $Ax = b$ is under-determined if $m<n$ and has infinite solutions for $v$." The author should provide a more explicit explanation about how we can formally guarantee independence when $A$ is a neural network since, at least it is not very intuitive to the reviewer.**
>
> Thanks for your careful reading. First, we wish to emphasize that $A$ is not the neural network but the input matrix of the linear layer. Each row of the original input matrix A is not necessarily independent. So the Gaussian elimination was used (line 11 in our algorithm 1) to guarantee that e
> ach non-zero row of input matrix A is independent. Thus, after Gaussian elimination, the condition that the rows in $A$ are independent is satisfied and we can use SOS to change the parameters of our neural network.
>
> >**Q2. The author considers the neural network a linear system, which is fine for the linear layers in the neural network when we omit the non-linear activation function. However, as modeled in Equation (3), the neural network is a stack of linear layers with corresponding nonlinear activation functions. How will the exact choice of the nonlinear activation function have an impact on the theoretical results?**
>
> We wish to clarify that the nonlinear activation function has no impact on our method.
> The linear lay with activation can be represented as:
> \begin{equation}
>     \sigma(Av + b) = c
> \end{equation}
> In our method, the input $A$ and output $Av+b$ are both fixed, we only change the parameters $v$ to $v^*$ such that the output is unchanged.
> In this case, no matter what the activation function is, the output of the linear layer will also not be changed.
>
> >**Q3. From Tables 1 and 2 we can observe that the SOS performs the best when it integrates with the Mixup. Can the author provide any interpretation and reasoning about this phenomenon? Why the basic training strategy can not bring sufficient benefit for the proposed method?**
>
> The data augmentation strategy Mixup will increase the accuracy of the baseline. Our SOS is also compatible with those data augmentation strategies and thus can further improve the test accuracy. This phenomenon also appears in other sharpness minimization methods [1, 2].
>
> >**Q4. The improvement in the object detection task in Table 4 is too marginal. It is better for the author to include the mean and std for this table and also for other tables.**
>
> Thanks for the suggestion, we have repeated the object detection experiment three times and added the mean and std for Table 4 in the revised version. As it takes time to repeat all experiments, the mean and std for other tables will be added in the final version.
>
> | VOC | Yolo-V5s | Yolo-V5x |
> | ----------- | ----------- | -------|
> | mAP w/ SOS  | $83.4_{\pm 0.2}$  | $87.1_{\pm 0.2}$|
> | mAP w/o SOS | $83.7_{\pm 0.1}$ |$87.4_{\pm 0.2}$ |
>
> **References:**
> * [1] Pierre Foret, Ariel Kleiner, Hossein Mobahi, and Behnam Neyshabur. Sharpness-aware minimization for efficiently improving generalization. In ICLR, 2021
> * [2] Yang Zhao, Hao Zhang, and Xiuyuan Hu. Penalizing gradient norm for efficiently improving gen-
> eralization in deep learning. In ICML, 2022.

---

> > ### Comment · Reviewer_KEY9 · 2023-11-19
> >
> > Thanks to the author for the precise response, the reviewer is more convinced with the proposed method. This is a good paper and deserves to be seen by the community. I raised my score to 8.

---

> > > ### Author Response · Authors · 2023-11-22
> > >
> > > Thank you very much for your constructive suggestions and positive rating!
> > >
> > > Best,
> > >
> > >  Authors

---

### Official Review · Reviewer_urCd · 2023-11-01

**Soundness:** 3 good
**Presentation:** 3 good
**Contribution:** 2 fair
**Rating:** 5
**Confidence:** 4

**Summary:**

This paper introduces optimum shifting (OS) to change the parameters of a neural network from sharper minima to a flatter one, while maintaining the same training loss. The authors prove that the minima got by their method is actually flatter than the original one. Furthermore they introduce a stochastic optimum shifting (SOS) utilizing neural collapse theory for practical implementations, which reduces computational costs and provide more degrees of freedom for optimum shifting.

**Strengths:**

The paper is clearly written and the idea is easy to follow.

The proposed OS method is easy to understand and easy to implement.

The SOS technique seems to bring improvement to multiple tasks.

**Weaknesses:**

1. The SOS which uses batched data instead of whole dataset for training relies on the neural collapse phenomenon, but neural collapse just happen in the terminal phase of training.  This means before the stage of getting 0 training error, the rely on neural collapse is not valid. Hence the assumption that "if the loss of 100 images remains unchanged ... the loss of the entire dataset will also remain nearly unchanged" will not hold for the stage of training before 0 error.

2. The empirical results are only on image tasks and this makes the claim of improvement by SOS not very convincing.

**Questions:**

Could you share your thoughts regarding weakness 1 (see above)?

---

> ### Author Response · Authors · 2023-11-14
>
> We appreciate the constructive comments on our paper. The raised concerns are addressed as follows.
> > **Q1. The SOS which uses batched data instead of
> whole dataset for training relies on the neural col-
> lapse phenomenon, but neural collapse just happen
> in the terminal phase of training. This means before
> the stage of getting 0 training error, the rely on neural collapse is not valid. Hence the assumption that
> ”if the loss of 100 images remains unchanged ... the
> loss of the entire dataset will also remain nearly unchanged” will not hold for the stage of training before
> 0 error.**
>
> Thank you for the good question. The stochastic OS is inspired by the NC phenomenon to keep the loss unchanged on a small batch. Rigorously speaking, we can keep the loss unchanged on CIFAR100 with 100 samples in the latter training stage. However, in the mid and beginning training phase, the data points will not totally converge to the class mean. So we use more samples (300 in the experiments for CIFAR100) and hope it can also keep the loss unchanged in the mid and beginning training phase. Moreover,  from the experimental view (Figure 2 of our paper), we can see that the loss with stochastic OS is almost the same or lower than the loss without stochastic OS, which further validates our statement that stochastic OS will not affect the training loss value in the whole training process.
>
> > **Q2. The empirical results are only on image tasks and this makes the claim of improvement by SOS not very convincing.**
>
> For other tasks such as NLP, the NC phenomenon may not exist and it needs other theory to guarantee the loss unchanged after SOS. So in this paper, we mainly focus on image tasks as in Foret et al [1], Kwon et al. [2] and Zhao et al. [3]. Thanks for the suggestion, the research of our SOS on other tasks is planned for our future work.
>
> **References:**
> * [1] Pierre Foret, Ariel Kleiner, Hossein Mobahi, and Behnam Neyshabur. Sharpness-aware minimization for efficiently improving generalization. In ICLR, 2021
> * [2] Jungmin Kwon, Jeongseop Kim, Hyunseo Park, and In Kwon Choi. Asam: Adaptive sharpness aware minimization for scale-invariant learning of deep neural networks. In ICML, 2021.
> * [3] Yang Zhao, Hao Zhang, and Xiuyuan Hu. Penalizing gradient norm for efficiently improving gen-
> eralization in deep learning. In ICML, 2022.

---

> ### Author Response · Authors · 2023-11-21
> **Kind Reminder**
>
> Dear Reviewer,
>
> Thank you again for your great efforts and valuable comments in reviewing our paper. We have carefully addressed the main concerns in detail in the rebuttal and we hope you might find the response satisfactory. As the discussion phase is about to close, we are very much looking forward to hearing from you about any further feedback. We will be very happy to clarify further concerns (if any).
>
> Best,
>
> Authors

---

### Official Review · Reviewer_DgxS · 2023-11-01

**Soundness:** 2 fair
**Presentation:** 3 good
**Contribution:** 2 fair
**Rating:** 3
**Confidence:** 3

**Summary:**

Authors proposed an efficient method to improve the generalization of trained DNN models via stochastic optimum shifting on a small sample set leveraging neural collapse.

**Strengths:**

Authors provided thorough proof on the theorems. This method seems to have marginal improvements over benchmark results using a small set of data to improve generalization. If there is a use case for this, it's a promising experiment.

**Weaknesses:**

As the authors mentioned. "there always exist models with good generalization but with arbitrarily large sharpness". I question the contribution made here is significant enough.

**Questions:**

why focus on generalization techniques after model is trained and not on producing a flatter minima during training. I think this is good after the fact but the effort is better spent on how to improve generalization during training.

---

> ### Author Response · Authors · 2023-11-14
>
> We appreciate the constructive comments on our paper. The raised concerns are addressed as follows.
>
> > **Q1. As the authors mentioned.** **"there always exist models with good generalization but with arbitrarily large sharpness".** **I question the contribution made here is significant enough.**
>
> Thanks for the comment. As lots of recent studies, such as Foret et al [1]; Kwon et al. [3] and Zhao et al. [4], propose to penalize the sharpness of the landscape for improving the generalization, the contribution to minimizing the sharpness of the loss
> landscape is of vital importance.
>
> Moreover, theoretically, as we stated in line 7 in our related work, although there always exist models
> with good generalization but with arbitrarily large sharpness, it does not contradict our main result and contribution here, which only asserts the interpolation solution with a minimal trace of Hessian generalizes well, but not vice versa. This is also stated and
> supported by Gatmiry et al. [2].
>
> Our contribution is also acknowledged by Reviewer KEY9 in the reply: This is a good paper and deserves to be seen by the community.
>
> > **Q2. Why focus on generalization techniques after model is trained and not on producing a flatter minima during training. I think this is good after the fact but the effort is better spent on how to improve generalization during training.**
>
> Thanks for your useful advice. Our experiment section focuses on both the trained model and during training process. We apply SOS to the trained model and during training separately in Sec. 5.1 and in Sec. 5.2. In Sec. 5.2, we present a thorough analysis of applying SOS during training, including loss, accuracy, and model weight analysis. Like Foret et al [1]; Kwon et al. [3] and Zhao et al. [4], we mainly focus on improving the test accuracy on image classification tasks on different models during training.  Moreover, we have also done experiment on object detection task and added the Hessian analysis during training in the Appendix of our updated paper for further verification. We thank your good suggestions again.
>
> **References:**
> * [1] Pierre Foret, Ariel Kleiner, Hossein Mobahi, and Behnam Neyshabur. Sharpness-aware minimization for efficiently improving generalization. In ICLR, 2021
> * [2] Khashayar Gatmiry, Zhiyuan Li, Ching-Yao
> Chuang, Sashank Reddi, Tengyu Ma, and Stefanie Jegelka. The inductive bias of flatness regularization for deep matrix factorization. arXiv preprint arXiv:2306.13239, 2023.
> * [3] Jungmin Kwon, Jeongseop Kim, Hyunseo Park, and In Kwon Choi. Asam: Adaptive sharpness aware minimization for scale-invariant learning of deep neural networks. In ICML, 2021.
> * [4] Yang Zhao, Hao Zhang, and Xiuyuan Hu. Penalizing gradient norm for efficiently improving gen-
> eralization in deep learning. In ICML, 2022.

---

> ### Author Response · Authors · 2023-11-21
> **Kind Reminder**
>
> Dear Reviewer,
>
> Thank you again for your great efforts and valuable comments in reviewing our paper. We have carefully addressed the main concerns in detail in the rebuttal and we hope you might find the response satisfactory. As the discussion phase is about to close, we are very much looking forward to hearing from you about any further feedback. We will be very happy to clarify further concerns (if any).
>
> Best,
>
> Authors

---

### Meta-Review · Area_Chair_6we6 · 2023-12-02

**Metareview:**

This paper builds on the idea of shifting the weights of a model to flat local minima. The goal is to let the neural network teleport to a point where the training loss remains the same while flatness, measured by the trace of Hessian with respect to the weights, is reduced. The proposed method is based on minimizing the weight-norm of the FC layer. To reduce the complexity of finding the solution that preserves the training loss on all samples, the proposed method instead keeps the loss unchanged on a batch of samples. The final results demonstrate some improvement in the generalizability of the trained models on a few standard CV benchmarks.

**Justification For Why Not Higher Score:**

There are many loose ends in the paper.
(1) I don't think the theorems are fully fleshed out. Some proof steps are skipped, and the theorems are weak because the constants C0/C1/C2, etc., involve the trace of non-interpretable data-dependent quantities. The flatness regularization effect depends greatly on the specific form of the connection between the trace and ||v||^2, but the theorems can not shed light on these connections.
(2) The connection to neural collapse needs more clarification. It is worth highlighting more discussions on neural collapse and more empirical measurements of neural collapse.
(3) I don't understand how mixup comes into the picture.
(4) The effect of flatness reduction (in Figure 3) has no baselines. It's hard to tell how significant the regularization effect is.

**Justification For Why Not Lower Score:**

N/A

---

### Decision · Program_Chairs · 2024-01-16

Reject